# Reinforcement Learning for Versatile Video Reasoning Capabilities in Base Multimodal LLMs

## Abstract

Multimodal Large Language Models (MLLMs) have made great progress in video understanding tasks. However, when it comes to understanding complex or lengthy videos, MLLMs tend to overlook details or produce hallucinations. To alleviate these issues, recent work has attempted to leverage reinforcement learning (RL) to boost models' deep linguistic reasoning of complex videos. But these methods have two main problems: First, the RL framework they used has unstable training, high training costs, and is difficult to train satisfactory video reasoning models; Second, the linguistic reasoning process is difficult to guarantee the reliability of visual information. To alleviate these problems, we propose to use multimodal elements for reasoning, and we design a novel framework to build and enhance versatile video reasoning capabilities on MLLMs. We carefully design a multi-task cold start and multi-task reinforcement learning to improve the model's visual perception and proficiency in multiple capabilities. In the inference phase, we leverage multimodal reasoning and dynamic sampling to further improve the performance. We verified the efficiency of the framework on a base MLLM (Qwen2-VL-7B-Base). Through cold-start with 3k data and reinforcement learning training with 5k data, combined with inference design, our final model significantly outperforms the base model on seven public video benchmarks, even surpassing and approaching the state-of-the-art Instruct Models such as Qwen2.5-VL-7B-Instruct.

## 1 Introduction

The recent surge of large language reasoning models (Jaech et al., 2024; Shao et al., 2024; Guo et al., 2025a; Yang et al., 2025a) has marked great progress towards a new era of artificial intelligence, particularly in addressing challenging and realistic tasks such as mathematics, reasoning, etc. These advances have also promoted the rapid development of multimodal large language models (MLLMs) (OpenAI, 2025; Hurst et al., 2024a; Guo et al., 2025b). A notable trend is the extension of reinforcement learning (RL) methods from the linguistic to the multimodal domain. Recent studies focus on improving RL algorithms (Meng et al., 2025), designing more effective verifiers (Wang et al., 2025b;d), and expanding to diverse modalities (Wang & Peng, 2025; Zhao et al., 2025a).

Despite the promising performance of MLLMs trained with RL algorithms on image understanding tasks (Chen et al., 2025a; Zhang et al., 2025b), RL training on video tasks has not achieved comparable improvements (Feng et al., 2025; Wang et al., 2025a), and establishing a stable and efficient RL framework for video understanding remains an open challenge. Existing RL training frameworks (Feng et al., 2025; Chen et al., 2025b) for videos typically involve high-cost long chain-of-thought (CoT) reasoning annotations, large-scale CoT cold start, and large-scale RL training. For example, Video-R1 (Feng et al., 2025) curates 165k data for cold start and 260k data for RL training. These models build on Instruct Models (e.g., Qwen2.5-VL-Instruct), which have already undergone large-scale SFT and post-training on image and video tasks using a direct-response paradigm, i.e., generating a brief answer immediately in response to a question. However, the inherent prior of such models favors direct responses over step-by-step reasoning, limiting their performance on tasks requiring logical reasoning (Zhang et al., 2024b). Therefore, it requires large-scale data and extensive training to enable the model to develop robust reasoning capabilities and adapt to reasoning formats and tasks. Although recent methods (Feng et al., 2025; Wang et al., 2025a) have enhanced the model's

ability to reason over video content and generate summaries, they still lag behind the original Instruct Models, revealing a performance gap between reasoning training and direct-response models.

Furthermore, recent works (Feng et al., 2025; Guo et al., 2025b) only use RL training to incentivize the linguistic reasoning path, which may include reasoning contents such as problem analysis, video perception, information reasoning, and summary. However, using only a textual reasoning path makes it difficult to ensure the long-term accuracy of visual information, and long-term text reasoning is prone to lead to incorrect and hallucination (Lanham et al., 2023). For video tasks, more effective and accurate reasoning paradigms involving multimodal elements should be explored instead of relying merely on text elements, in order to better cope with reasoning-intensive video understanding tasks. Meanwhile, although giving the model a longer budget during reasoning (e.g., more than 1k tokens) can stimulate a self-reflective mechanism during the reasoning process, it also leads to a longer inference time, which will become a crucial bottleneck in real-world video applications.

To address these limitations, we propose **VideoReasoner**, an efficient framework that builds and enhances versatile video reasoning capabilities for base MLLMs. To construct a simple and usable framework and verify the effectiveness of this framework. We set the task to conduct training based on Base MLLMs. The basic goal is to use this framework to enhance video understanding capabilities more efficiently; that is, the video understanding performance needs to exceed the corresponding Instruct Model, and explore whether this framework can bring about effects beyond expectations. As mentioned earlier, there is an inherent performance gap when using the Instruct Model. The usage of the Base Model is proposed here to avoid the performance gap and to verify the effectiveness of the basic framework. However, in the experimental section, we also presented the training results based on the Instruct Model. As for how to bridge this gap, more subsequent work is needed for exploration. Meanwhile, since this framework subsequently designs the reasoning process of multimodal elements, which involves multitask learning, the Base Model is more suitable as a baseline because it has only undergone multimodal pre-training and can adapt to multitask learning more efficiently.

To avoid error accumulation and hallucinations of visual content understanding caused by only using textual reasoning, we extend it to three perspectives: event reasoning, keyframe reasoning, and direct-response. As important contents in videos, events and keyframes can express clearer information than text. To this end, we design a two-stage training method to enable the model to learn and enhance its reasoning ability from these perspectives. In the first stage, a multi-task SFT is designed as a cold start. We design a unified instruction for multiple tasks as shown in Table 6. These tasks have different task prefixes and subsequent specific contents. An example is shown in the left part of Figure 1. For keyframe reasoning, we do not directly output the indices of keyframes for reasoning. Instead, we adopt a simple approach that predicts the key elements. In the second stage, we propose a novel multi-task reinforcement learning method. For a given video-question pair, multiple sets of responses are generated using prompts with different task prefixes, and task-specific rewards are defined for Group Relative Policy Optimization (GRPO). After training, we design an inference pipeline leveraging the model's three video reasoning capabilities. Specifically, the model performs event reasoning and keyframe reasoning in parallel, conducts dense sampling of the outputs, and then feeds the sampled video frames back into the model to generate a direct response.

Through extensive experiments on various public video benchmarks, including general video understanding, video reasoning, and video temporal grounding benchmarks, we validate the effectiveness of the proposed VideoReasoner framework. Based on Qwen2-VL-7B-Base, through a cold start with 3k samples and RL training with 5k samples, combined with the proposed inference pipeline, the framework achieves substantial improvements across seven benchmarks, outperforming Qwen2-VL-7B-Instruct on five benchmarks and Qwen2.5-VL-7B-Instruct on three benchmarks. Compared with the data and training costs required for SFT or post-training used in training two Instruct Models, our framework requires only 8k data while achieving comparable results, demonstrating its efficiency and highlighting its potential for real-world applications.

## 2 RELATED WORK

### 2.1 MULTIMODAL LARGE LANGUAGE MODELS FOR VIDEOS

Multimodal Large Language Models (MLLMs) (Hurst et al., 2024b; Anthropic, 2024; Comanici et al., 2025; ByteDance, 2025; Zhu et al., 2025; Bai et al., 2025) have achieved significant advancements in

video understanding tasks, and open-source models are gradually catching up with closed-source models in terms of multimodal capabilities. MLLMs treat video input as a sequence of images and bridge the visual tokens and language space through a modality alignment module, and these works use Q-Former (Li et al., 2023) to aggregate temporal information or simple MLP projectors. The training paradigms of MLLMs for video understanding continue to evolve. Recently, Qwen2.5-VL (Bai et al., 2025) fuses adjacent frames and further compresses encoded multiple visual tokens into a single token, which is then connected to the language model via MLP. To enhance temporal awareness, some works propose explicit temporal textual prompts (Ren et al., 2024), temporal module (Zeng et al., 2025), and MRoPE techniques (Bai et al., 2025). As for video training, many works adopt a hybrid data training strategy. For example, InternVL2.5 (Chen et al., 2024b) and Qwen2.5-VL are trained on a combination of single images, multi-frame image sequences, and videos. Additionally, post-training techniques are widely used to improve video reasoning performance (Bai et al., 2025; Zhu et al., 2025; Guo et al., 2025b). In parallel, video benchmarks have been introduced to assess the various MLLMs, such as general video understanding tasks (Fu et al., 2024; Wang et al., 2024b; Wu et al., 2024) and video reasoning tasks (Yang et al., 2024; Zhao et al., 2025b). Recently, the use of reinforcement learning to enhance the reasoning ability of models (Wang & Peng, 2025; Feng et al., 2025), the ability to use tools (Zhang et al., 2025a), and evolve into video agents (Zhang et al., 2025c) are the cutting-edge directions for the development of multimodal large language models towards more powerful and practical video understanding.

## 2.2 Multimodal Reasoning Large Language Models

Large language reasoning models using Chain-of-Thought (CoT) (Wei et al., 2022), test-time scaling (Jaech et al., 2024), and reinforcement learning (RL) (Shao et al., 2024) have achieved great success for the reasoning and instruction-following abilities, such as in mathematics, coding, and agentic tasks. Recently, DeepSeek-R1 (Shao et al., 2024) demonstrates that large-scale RL with verifiable rewards induces emerging reasoning capabilities in LLMs. Inspired by this, the introduction of reasoning and design reasoning on multimodal large models (MLLM) is continuously evolving and has demonstrated some promising results (Xu et al., 2024a; Yang et al., 2025b; Liu et al., 2025; Peng et al., 2025; Wang et al., 2025b). To explore the multimodal reasoning effect for complex video understanding tasks, some works focus on step-by-step reasoning, CoT training, and RL training. VideoCoT (Wang et al., 2024c) propose a high-quality video dataset with chain-of-thought reasoning annotations. Video-of-Thought (Fei et al., 2024) breaks down a complex video task into simpler sub-problems, such as tracking or action analysis, and it addresses them using step-by-step reasoning from a low-level pixel perception to high-level cognitive interpretation. For Video-MLLM RL training, (Wang & Peng, 2025; Feng et al., 2025) introduce GRPO training for MLLM to reasoning for fully understanding the video-language relationship before the final answer. Recent RL training frameworks often involve offline and online training stages. Seed1.5-VL (Guo et al., 2025b) incorporates video data into the pretraining phase and designs a post-training phase through a combination of supervised fine-tuning (SFT) and RL techniques. Keye-VL-1.5 leverages a slow-fast video encoding method, and the post-training stage is continuous iterative SFT and RL training. InternVL3.5 (Wang et al., 2025c) also propose a cascade RL framework that consists of a mixed preference optimization and an online RL stage. However, these RL frameworks merely rely on linguistic reasoning to analyze and interpret video content. While our proposed framework involves not only linguistic reasoning but also multimodal element reasoning.

## 3 Method

**Overview** In this section, we propose a two-stage training framework to build and enhance versatile video capabilities for base Multimodal Large Language Models (MLLMs). The main idea is to fully explore the various video capabilities to enhance video understanding, with the expectation that the model can effectively leverage its pre-trained knowledge to perform these tasks. The framework consists of three steps: (1) a multi-task cold start (Section 3.2); (2) a multi-task RL (Section 3.3), and (3) an efficient video inference pipeline (Section 3.4). See Figure 1 for the overview.

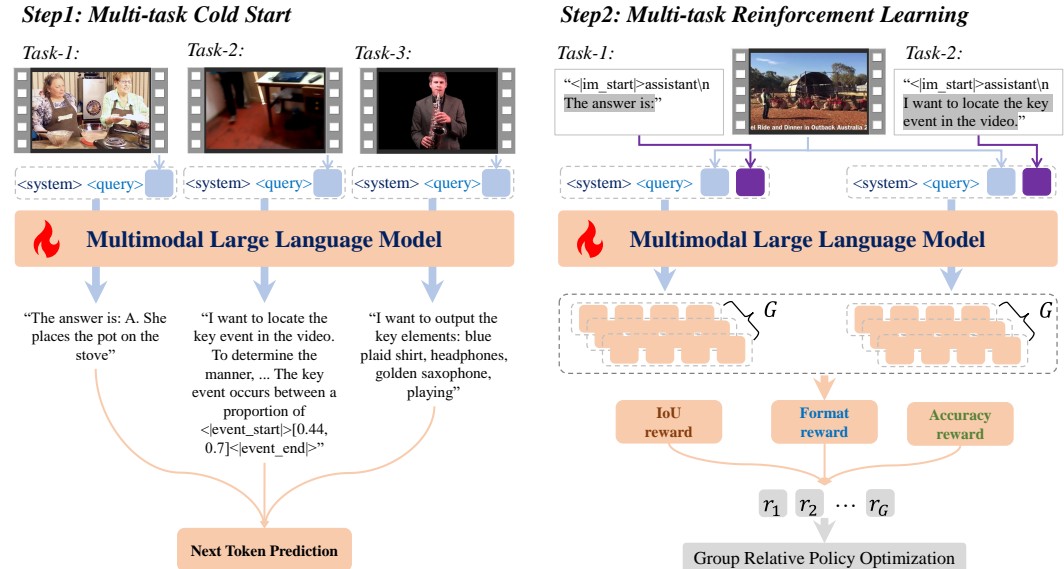

Figure 1: The proposed training paradigm aims to build and enhance versatile video reasoning capabilities for MLLMs, including a multi-task Cold Start and a multi-task RL with high efficiency.

## 3.1 BACKGROUND OF GRPO

Group Relative Policy Optimization (GRPO) (Shao et al., 2024) is proposed to save the training costs of reinforcement learning. It foregoes the critic model that is typically the same size as the policy model, and estimates the baseline from group scores instead. Specifically, for each question $q$, GRPO samples a group of outputs $\{o_1, o_2, \cdots, o_G\}$ from the old policy $\pi_{\theta_{old}}$ and then optimizes the policy model $\pi_\theta$ by maximizing the following objective:

$$\mathcal{J}_{GRPO}(\theta) = \mathbb{E}[q \sim P(Q), \{o_i\}_{i=1}^G \sim \pi_{\theta_{old}}(O|q)]$$

$$\frac{1}{G} \sum_{i=1}^G \left( \min\left( \frac{\pi_\theta(o_i|q)}{\pi_{\theta_{old}}(o_i|q)} A_i, \text{clip}\left( \frac{\pi_\theta(o_i|q)}{\pi_{\theta_{old}}(o_i|q)}, 1-\epsilon, 1+\epsilon \right) A_i \right) - \beta \mathbb{D}_{KL}\left( \pi_\theta || \pi_{ref} \right) \right), \tag{1}$$

$$\mathbb{D}_{KL}\left( \pi_\theta || \pi_{ref} \right) = \frac{\pi_{ref}(o_i|q)}{\pi_\theta(o_i|q)} - \log \frac{\pi_{ref}(o_i|q)}{\pi_\theta(o_i|q)} - 1, \tag{2}$$

where $\epsilon$ and $\beta$ are hyper-parameters, and $A_i$ is the advantage, computed using a group of rewards $\{r_1, r_2, \ldots, r_G\}$ corresponding to the outputs within each group:

$$A_i = \frac{r_i - \text{mean}(\{r_1, r_2, \cdots, r_G\})}{std(\{r_1, r_2, \cdots, r_G\})}. \tag{3}$$

## 3.2 MULTI-TASK SUPERVISED FINE-TUNING AS A COLD START

A base multimodal large language model (MLLM) is built upon a large language model and trained with multimodal pretraining to align visual signals with language tokens, such as Qwen2-VL (Wang et al., 2024a), which processes 1.4 trillion tokens during pretraining. Considering that the base model is exposed to large-scale and diverse linguistic and visual scenarios during the pre-training stage, it is expected that the model can utilize this prior knowledge for various visual tasks, including complex video understanding. To this end, we innovatively design three core tasks for the MLLM to learn: video question answering, video event grounding, and keyframe detection. However, "keyframes" are difficult to define, and current MLLMs still struggle to predict them accurately. To address this, we reformulate keyframe detection as key element generation and employ a visual encoder to retrieve the corresponding video frames.

To support the model's adaptability to the three aforementioned tasks, we curate datasets specifically tailored for each task. For video question answering, which is the most commonly used task for training MLLMs, we adopt multi-choice QA data from the training sets of (Feng et al., 2025). For video event grounding, we use the temporal grounding training set from (Gao et al., 2017). For key element generation, we collect raw videos from (Zhang et al., 2024c) and employ a proprietary model (Guo et al., 2025b) to generate key elements, constructing paired video and textual key element annotations. For each input video $X_v$ and instruction (comprising a system prompt and a video query), the target answer corresponds to one of three types, as illustrated in the left part of Figure 1. The most distinctive feature of these three responses lies in their prefixes: "The answer is:", "I want to locate the key event in the video.", and "I want to output the key elements:". All tasks share the same system prompt, which specifies the principles the model should follow, as detailed in Table 6.

For a sequence of length $L$, the probability of various target answers is defined as follows:

$$p(X_a|X_v, X_{instruct}) = \prod_{i=1}^{L} \pi_\theta(x_i|X_v, X_{instruct,<i}, X_{a,<i}) \tag{4}$$

where $\theta$ is the trainable parameter of MLLM. All parameters of the visual encoder and the language model backbone are trainable. $X_{instruct,<i}$ and $X_{a,<i}$ are instruction and answer tokens before the current prediction token $x_i$, and the whole response (including the prefix) is used to compute the loss for next token prediction.

For video event grounding, we observe that using absolute numerical predictions (Zeng et al., 2025) makes the model's outputs highly dependent on the training distribution. For instance, if the model is trained on short videos, it tends to predict very small time values and has difficulty handling event localization in long videos. To address this issue, we propose a relative numerical prediction for event grounding, i.e., predicting the time ratio for durations [start_ratio, end_ratio]. We also insert two learnable special tokens <|event_start|> and <|event_end|> to make the model stably predict the grounding results when performing this task.

We employ only pre-trained multimodal large language models for multi-task fine-tuning instead of adopting post-trained SFT models. Although the latter typically achieve stronger performance, they also exhibit stronger preferences or inductive biases and demand larger-scale multi-task datasets for effective fine-tuning. In our setting, we utilize approximately 3k training samples in total, with each task accounting for around 1k samples.

### 3.3 MULTI-TASK REINFORCEMENT LEARNING

The Cold Start enables the base MLLM to adapt to the responses of various tasks, but it is more about proficient format output rather than truly effective learning of these capabilities. To enhance these capabilities, we utilize the commonly used reinforcement learning algorithm GRPO (Shao et al., 2024) to incentivize the diverse capabilities of the model. Although recently there have been some works focusing on RL for video understanding (Feng et al., 2025; Wang et al., 2025e), they only use GRPO for a single task, such as video QA or temporal grounding. In this work, we propose a novel multi-task GRPO algorithm. At the model level, the model can generate rollouts of multiple tasks and separate optimized policy models based on the relative advantages between groups obtained from the rollouts of different tasks. At the data level, given the same video-query pair, we use different prefix hints to prompt the model to roll out different tasks for the same query, effectively improving the utilization efficiency of data.

As shown in the right part of Figure 1, we select two tasks—event grounding and video QA—to construct the multi-task GRPO for video training. Given the same video and query, we prepend task-specific prefixes after the default assistant generation prompt (e.g., <assistant>). Specifically, the event grounding prefix is: "I want to locate the key event in the video.", and the video QA prefix is: "The answer is:". Importantly, we do not need to manually construct such data. The dataset of (Xiao et al., 2024) provides both the answer metadata and reference time intervals for the same video-query pairs. Although (Xiao et al., 2024) introduces a new video QA task, we instead repurpose this dataset to build the training set for multi-task GRPO.

Unlike the Cold Start stage, where the model is required to predict task prefixes, in GRPO, the tokens corresponding to task prefixes are not used to compute the loss. Given a video $X_v$, and a query $q$,

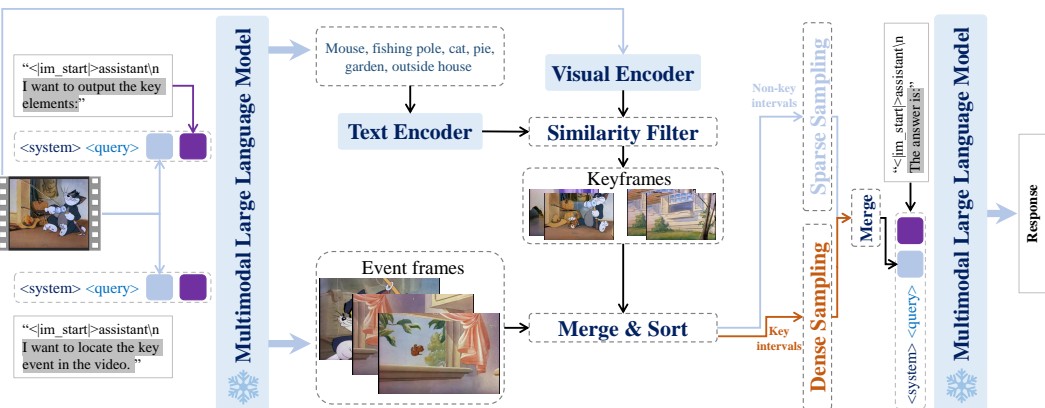

<query> : Question: From where does the cartoon mouse escape to the outside of the house?\nOptions:\nA. Door\nB. Tunnel\nC. Hole in the wall\nD. Window

Figure 2: Inference for video understanding using three capabilities of the enhanced model.

generation prompt $g$, and a task prefix in $\{p_1, p_2\}$, a multimodal query is defined as $m = [X_v, q, g, p]$, where $[\cdot]$ denotes token concatenation. The objective of multi-task GRPO is defined as:

$$\mathcal{J}_{M\text{-}GRPO}(\theta) = \mathbb{E}[p \sim \{p_1, p_2\}, \{o_i\}_{i=1}^G \sim \pi_{\theta_{old}}(O|m)\frac{1}{G}\sum_{i=1}^G \left[ \min\left(\frac{\pi_\theta(o_i|m)}{\pi_{\theta_{old}}(o_i|m)}A_i, \right.\right.$$

$$\left.\left. \text{clip}\left(\frac{\pi_\theta(o_i|m)}{\pi_{\theta_{old}}(o_i|m)}, 1-\epsilon, 1+\epsilon\right)A_i\right) - \beta\mathbb{D}_{KL}\left(\pi_\theta||\pi_{ref}\right)\right] \tag{5}$$

where KL divergency $\mathbb{D}_{KL}$ and advantage $A_i$ are are the same as defined in Equation 2 and Equation 3, respectively.

**Reward Modeling**   The definition of the reward $r_i$ guides the model's learning objective. Our goal is to enable the model to find the direction for optimization within its own search space while maintaining proficiency in multiple tasks. At the same time, since we choose not to predict task prefixes, we prevent the model from getting stuck on which format to output or overfitting to a single format during rollout and optimization. Instead, it performs autoregressively to continue generating subsequent tokens for a given specific prefix. To better leverage the role of the two tasks in enhancing video comprehension ability, we have designed three rewards, including an IoU reward $r_{\text{IoU}}$, a format reward $r_{\text{form}}$, and an accuracy reward $r_{\text{acc}}$.

**IoU Reward** $r_{\text{IoU}}$: this reward is designed for the event grounding task. It is computed as the IoU between the predicted interval ratio and the ground-truth interval ratio.

**Format Reward** $r_{\text{form}}$: this reward is used for the event grounding task to evaluate whether the model correctly predicts the two special tokens, <|event_start|> and <|event_end|>.

**Accuracy Reward** $r_{\text{acc}}$: this reward is for the video QA task, and it takes a value of either 0 or 1.

For the collaborative optimization of the two tasks, GRPO only computes importance weights of the rollout tokens, targeting the probability of the model's response to the input sequence. Thus, although query data from different tasks are sampled in the same batch, the optimization objectives of each task do not interfere with each other. For the event grounding task, the objective is to optimize the model's response quality given the input video, query, and task prefix. In this case, the accuracy reward can be set to 0, allowing the model to focus on improving the format reward $r_{\text{form}}$ and IoU reward $r_{\text{IoU}}$. For the Video QA task, the goal is to optimize the model's response quality for this task input. We set the format reward and IoU reward to 0 and let the model improve the accuracy reward $r_{\text{acc}}$. The overall reward function is defined as their sum:

$$r(o) = r_{\text{IoU}} + r_{\text{form}} + r_{\text{acc}} \tag{6}$$

For the training data of this stage, we use only 5k video queries. The training at this stage is very data-efficient and does not require long context rollouts, resulting in high training efficiency.

Table 1: Performance on public video benchmarks compared to previous models.

| Model \Benchmark | Video-MME | | | | LongVB | MLVU | LVBench | VideoEval-Pro |
|---|---|---|---|---|---|---|---|---|
| | Short | Medium | Long | Overall | Val | M-Avg | Overall | MCQ |
| LLaVA-Video-72B (Zhang et al., 2024c) | 81.4 | 68.9 | 61.5 | 70.6 | 62.4 | 71.3 | 46.1 | 50.1 |
| InterIVL2.5-72B (Chen et al., 2024b) | 82.8 | 70.9 | 62.6 | 72.1 | - | - | 37.9 | - |
| PLLaVA-7B (Xu et al., 2024b) | - | - | - | - | 40.2 | - | - | - |
| LongVA (Zhang et al., 2024a) | 61.4 | 50.9 | 45.0 | 52.4 | - | 56.3 | - | 38.0 |
| Long-LLaVA (Song et al., 2024) | 61.9 | 51.4 | 45.4 | 52.9 | - | - | - | 36.9 |
| Kangaroo-8B (Liu et al., 2024) | 66.1 | 55.3 | 46.6 | 56.0 | 54.2 | - | - | - |
| VideoTree (Wang et al., 2025f) | - | - | - | 56.1 | 52.3 | - | - | - |
| InternVL2-8B (Chen et al., 2024c) | 68.0 | 52.0 | 48.9 | 56.3 | 54.6 | 56.3 | - | 39.9 |
| ViLA-1.5-8B (Lin et al., 2024) | - | - | - | 58.2 | 56.3 | 56.7 | - | - |
| Qwen2-VL-7B-Instruct (Wang et al., 2024a) | 70.7 | 57.6 | 50.2 | 59.3 | 55.2 | 61.7 | 39.7 | 39.6 |
| LongVILA (Chen et al., 2024a) | - | - | - | 60.1 | - | - | - | - |
| MiniCPM-V-2.6 (Yao et al., 2024) | 71.3 | 59.4 | 51.8 | 60.9 | 54.9 | - | - | - |
| LongVILA-R1 (Chen et al., 2025b) | - | - | - | **62.4** | - | - | - | - |
| **GRPO** | | | | | | | | |
| Baseline: Qwen2.5-VL7B-Instruct | **74.6** | **61.2** | 51.4 | **62.4** | **59.3** | 63.0 | 37.7 | 40.3 |
| + Video-R1 (Feng et al., 2025) | 72.2 | 59.4 | 47.0 | 59.5 | 49.7 | 62.0 | 38.6 | 42.2 |
| **Our Multi-task GRPO** | | | | | | | | |
| Baseline: Qwen2-VL-7B-Base | 68.3 | 57.0 | 49.6 | 58.3 | 53.7 | 61.4 | 36.8 | 39.2 |
| + RL | 72.1 | 58.1 | 52.3 | 60.8 | 53.9 | 63.0 | 38.4 | 41.0 |
| + VideoReasoner | 72.9 | 60.6 | **53.0** | 62.0 | 55.0 | **64.6** | 44.6 | **44.1** |
| Δ | +4.6↑ | +3.6↑ | +3.4↑ | +3.7↑ | +1.3↑ | +3.2↑ | +7.8↑ | +4.9↑ |

## 3.4 INFERENCE WITH VERSATILE VIDEO REASONING CAPABILITIES

After reinforcement learning, the event grounding and video understanding abilities are enhanced, and we try to combine the two abilities and keyframe detections to build an efficient video question-answering process. As shown in Figure 2, we first input the video along with two types of task prefixes to the MLLM, prompting the model to output the duration of the related event and key elements in parallel. For key elements, we use a text encoder (Bolya et al., 2025) to extract the textual embeddings, and feed the uniformly sampled video frames into a visual encoder (Bolya et al., 2025) to obtain video embeddings, from which keyframes with high similarity are selected.

Specifically, the event grounding process outputs the most important event interval ratio $[S_r, E_r]$, which is multiplied by the video duration to obtain the absolute time interval $[S_t, E_t]$. The keyframe detection process generates a series of keyframe positions. Based on the original segments division, we obtained a series of time segments $\{[S_{k_1}, E_{k_1}], [S_{k_2}, E_{k_2}], \cdots, [S_{k_n}, E_{k_n}]\}$. Then, we merge and sort all the selected time segments to obtain the key intervals. We adopt high fps dense sampling to fully utilize key visual information. Meanwhile, to prevent the model from ignoring global information, we also took into account other non-key time regions and implemented sparse sampling using a low fps. We merge and sort the sampled frames of these two parts, and input them together with the video QA prefix into the MLLM to obtain the final response.

## 4 EXPERIMENT

### 4.1 SETUP

Qwen2-VL-7B-Base (Wang et al., 2024a) is used as the base model, with all parameters of the MLLM fully fine-tuned. For analysis experiments, we also finetuned Qwen2-VL-7B-Instruct (Wang et al., 2024a) and Qwen2.5-VL-7B-Instruct (Bai et al., 2025). All experiments are conducted on H20 GPUs. We sample 64 frames per video for both training stages and inference. The 3k training data for the cold start stage are sampled from (Feng et al., 2025; Gao et al., 2017; Zhang et al., 2024c), and the 5k training data for RL training are sampled from (Xiao et al., 2024). Evaluation is performed on 7 video benchmarks: Video-MME (Fu et al., 2024), LongVB (Wu et al., 2024), MLVU (Zhou et al., 2024), LVBench (Wang et al., 2024b), VideoEval-Pro (Ma et al., 2025), VSI-Bench (Yang et al., 2024), and MMVU (Zhao et al., 2025b). The temporal grounding benchmark uses Charades-STA (Gao et al., 2017).

Table 2: Performance on public video reasoning benchmarks compared to previous models

| Model \Benchmark | VSI-Bench Overall | MMVU MCQ |
|---|---|---|
| LLaVA-OV-7B (Li et al., 2024) | 32.4 | 49.2 |
| ViLA-1.5-8B (Lin et al., 2024) | 28.9 | 49.2 |
| VideoTree (Wang et al., 2025f) | - | 54.2 |
| LLaVA-Video-7B (Zhang et al., 2024c) | 36.2 | 60.2 |
| Qwen2-VL-7B-Instruct (Wang et al., 2024a) | 33.4 | 63.4 |
| Baseline: Qwen2.5-VL-7B-Instruct (Bai et al., 2025) | **39.9** | **68.0** |
| + Video-R1 (Feng et al., 2025) | 37.8 | 64.3 |
| Baseline: Qwen2-VL-7B-Base (Wang et al., 2024a) | 28.9 | 61.1 |
| + Our RL | 33.7 | 62.4 |
| Δ | +4.8↑ | +1.3↑ |

Table 3: Performance on temporal grounding task.

| Model | Charades-STA (Gao et al., 2017) | | | |
|---|---|---|---|---|
| | mIoU | R1@0.3 | R1@0.5 | R1@0.7 |
| InternVideo2 (Wang et al., 2024d) | - | - | 70.0 | **48.9** |
| TimeSuite (Zeng et al., 2025) | - | 79.4 | 67.1 | 43.0 |
| Qwen2.5-VL-7B-Instruct (Bai et al., 2025) | 43.6 | 76.1 | 42.9 | 26.2 |
| **Ours** | | | | |
| - Cold Start | 54.1 | 78.6 | 62.4 | 35.8 |
| - RL | **59.1** | **81.9** | **70.4** | 45.7 |

## 4.2 MAIN RESULTS

**Quantitative Results**  Table 1 presents a comparison with previous models. Compared with current state-of-the-art models, our framework achieves the best or second-best performance on five benchmarks, except for LongVB. Notably, the results on MLVU, LVBench, and VideoEval-Pro surpass or are close to models with 72B parameters. The last row demonstrates that our framework achieves stable and significant improvements over the baseline. Our multi-task GRPO consistently improves accuracy across all benchmarks and tasks relative to our baseline, whereas the GRPO in Video-R1 exhibits decreases on three benchmarks and all tasks in Video-MME relative to its baseline. These results highlight the potential of our proposed framework for video understanding tasks.

Table 2 presents a comparison on video reasoning benchmarks. Qwen2.5-VL-7B-Instruct achieves the best result on both benchmarks. With the introduction of our multi-task RL training, the baseline improves by 4.8% on VSI-Bench and 1.3% on MMVU, surpassing Qwen2-VL-7B-Instruct on VSI-Bench. Compared with Video-R1, our multi-task RL approach shows consistent and stable performance improvements on video reasoning tasks.

Table 3 presents a comparison of temporal grounding on Charades-STA. Notably, although our baseline model, Qwen2-VL-7B, does not support temporal grounding, following cold-start training it outperforms Qwen2.5-VL-7B-Instruct. After RL training, our models achieve the best results in mIoU, R1@0.3, and R1@0.5, surpassing state-of-the-art models. These results demonstrate that our framework can develop and enhance the emerging temporal grounding capability of base MLLMs.

## 4.3 ABLATION STUDY

**Comparison of different settings in training**  Table 4 presents the detailed performance of cold start and RL training under single-task and two-tasks settings. First, the cold start consistently improves baseline performance. For RL training, using only temporal grounding data yields improvements only in the long-video scenario of Video-MME, while degrading performance on other tasks. The reason might be that this type of data does not use the final answer as the reward, which affects the accuracy of the model's response. Training using both video QA data and temporal grounding data simultaneously achieved better results than using only video QA data, demonstrating the effectiveness of multi-task RL training.

Table 4: Ablation Study for our proposed method.

| Model \Benchmark | Video-MME | | | | LongVB | MLVU | LVBench | VideoEval-Pro |
|---|---|---|---|---|---|---|---|---|
| | Short | Medium | Long | Overall | Val | M-Avg | Overall | MCQ |
| Baseline | 68.3 | 57.0 | 49.6 | 58.3 | 53.7 | 61.4 | 36.8 | 39.2 |
| - Cold Start | 70.4 | 58.1 | 50.2 | 59.6 | 53.8 | 61.1 | 38.0 | 40.3 |
| - RL training | | | | | | | | |
| w/ QA | 72.1 | 57.0 | 51.7 | 60.2 | 51.6 | 61.3 | 37.5 | 41.0 |
| w/ TG | 70.4 | 56.8 | 51.5 | 59.5 | 53.1 | 60.8 | 37.4 | 39.5 |
| w/ QA & TG | 72.1 | 58.1 | 52.3 | 60.8 | 53.9 | 63.0 | 38.4 | 41.0 |
| - Inference | | | | | | | | |
| w/ Event | 72.5 | 60.3 | 52.4 | 61.7 | 54.3 | 62.4 | 43.3 | 43.5 |
| w/ Keyframe | 72.6 | 58.9 | 52.4 | 61.3 | 54.8 | 63.4 | **44.6** | 43.0 |
| w/ Event & Keyframe | **72.9** | **60.6** | **53.0** | **62.0** | **55.0** | **64.6** | 42.8 | **44.1** |

Table 5: Analysis of multi-task Cold Start and multi-task RL training.

| Model \Benchmark | Video-MME | | | | LongVB | MLVU | LVBench | VideoEval-Pro |
|---|---|---|---|---|---|---|---|---|
| | Short | Medium | Long | Overall | Val | M-Avg | Overall | MCQ |
| Qwen2.5-VL-7B-Instruct | 74.6 | 61.2 | 51.4 | 62.4 | 59.3 | 63.0 | 37.7 | 40.3 |
| + Cold Start | 74.6 | 64.0 | 51.2 | 63.3 | 57.3 | 62.6 | 38.8 | 40.2 |
| + RL | 73.2 | 64.0↑ | 51.2 | 62.8↑ | 59.0 | 64.3↑ | 39.6↑ | 41.7↑ |
| Qwen2-VL-7B-Instruct | 70.7 | 57.6 | 50.2 | 59.3 | 55.2 | 61.7 | 39.7 | 39.6 |
| + Cold Start | 71.5 | 57.7 | 49.6 | 59.6 | 57.2 | 62.6 | 39.6 | 40.2 |
| + RL | 71.2↑ | 58.7↑ | 48.9 | 59.6↑ | 54.3 | 38.3 | 39.3 | 38.2 |
| Qwen2-VL-7B-Base | 68.3 | 57.0 | 49.6 | 58.3 | 53.7 | 61.4 | 36.8 | 39.2 |
| + Cold Start | 70.4 | 58.1 | 50.2 | 59.6 | 53.8 | 61.1 | 38.0 | 40.3 |
| + RL | 72.1↑ | 58.1↑ | 52.3↑ | 60.8↑ | 53.9↑ | 63.0↑ | 38.4↑ | 41.0↑ |

**Comparison of different settings in inference** The last three rows in Table 4 show the model's performance using different multimodal reasoning results during inference. Using either event grounding or keyframe detection results improves output accuracy, with comparable performance across benchmarks. Notably, combining both multimodal elements yields further gains, highlighting their complementary roles and importance in video reasoning.

**Comparison of different baselines in training** Table 5 presents the detailed training results of three baselines, including Qwen2.5-VL-7B-Instruct, Qwen2-VL-7B-Instruct/Base. As discussed in Sec 1, Instruct Models have stronger preference and bias than Base Models. To ensure a fair comparison, all three baselines are trained using the same dataset. Evaluation results are indicated with arrows, denoting metrics where the RL-trained model outperforms the baseline. When Qwen2.5-VL-7B-Instruct serves as the baseline, improvements are observed in 5 out of 8 indicators. With Qwen2-VL-7B-Instruct as the baseline, 3 out of 8 indicators improve, while a sharp decline is observed on MLVU, likely due to instruction disobedience after RL training. Using Qwen2-VL-7B-Base as the baseline, improvements are observed across all indicators. These results suggest that Base Models are more suitable for scalable RL training.

## 5 CONCLUSION

This work aims to establish a stable and efficient RL training framework for video understanding tasks. Unlike previous frameworks, which merely rely on linguistic reasoning for video content, we propose a novel framework that involves multimodal element reasoning, and our goal is to build and enhance versatile video reasoning capabilities on MLLMs. During the training phase, we proposed a multi-task cold start and a multi-task reinforcement learning. The collaboration of the two training stages can continuously improve the performance of the model and the ability of multimodal element reasoning. Based on the multimodal element reasoning capabilities, in the inference phase, we leverage multimodal reasoning and dynamic sampling to further improve the performance. We verified the efficiency of the proposed framework on a base MLLM. Through cold-start with 3k data and reinforcement learning training with 5k data, the final model significantly outperforms the base model on seven public video benchmarks, and even surpasses the state-of-the-art models trained by large-scale supervised fine-tuning.

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

## A    ETHICS STATEMENT

This study follows ethical guidelines and uses publicly available datasets and models, or those from published research. No human subjects were involved. There are no conflicts of interest or concerns regarding privacy, security, or discrimination. All research complies with applicable ethical standards and transparency requirements.

## B    REPRODUCIBILITY STATEMENT

The datasets and models used in this study are publicly available. Following the methods and experimental details outlined in the main text, we believe the results can be easily reproduced.

## C    USE OF LLMS

The originality of the research and the scientific contributions come solely from the authors, with no involvement of LLMs in the research tasks. No LLMs were used to write or revise the manuscript; the paper was written entirely by the authors. Automated tools were limited to standard utilities such as spell-checkers, citation managers, and LaTeX packages.

## D    DETAILS

```
System-message <STOP>
Human:
Given a video, please analyze the content carefully and provide
your response in one of the following formats:
    1.  **Event localization**:  locate the event using the format:
<|event_start|> [start_ratio, end_ratio] <|event_end|>, where the
ratios are floats between 0 and 1 indicating the relative position
in the video.
    2.  **Key elements extraction**:  list important elements or
actions in the video, output them as a comma-separated list:
    3.  **Direct answer**:  If the question can be answered directly
without additional processing, provide the answer clearly.
Question:  {Query} <STOP>
Assistant: X_a <STOP>
```

Table 6: The input sequence used to train base MLLM for multi-task, and only green sequence/tokens are used to compute the loss of the next token prediction.

## E    QUALITATIVE RESULTS

Figure 3 presents an inference example of the Needle Question Answering task. A video segment is embedded into a long video. It can be seen that uniform sampling is difficult to obtain the frames of these embedded short videos, which leads to incorrect responses from the MLLM. In our framework, apart from the model achieving capability enhancement through reinforcement learning, we propose to use a dynamic sampling method. For the results obtained from the event reasoning and keyframe reasoning of the model, we convert them into time intervals, using dense sampling in these intervals and sparse sampling in the remaining parts. Figure 4 presents an inference example of the Plot Question Answering task. Because the plot of this kind of video is very rich, when uniformly sampled video frames are input into an MLLM, the model has difficulty distinguishing important information, which may lead to hallucinations or errors in the responses. Our method first conducts event reasoning and keyframe reasoning. After dense sampling of the obtained results, important visual information can be directly input into the MLLM to improve the accuracy of the model's responses.

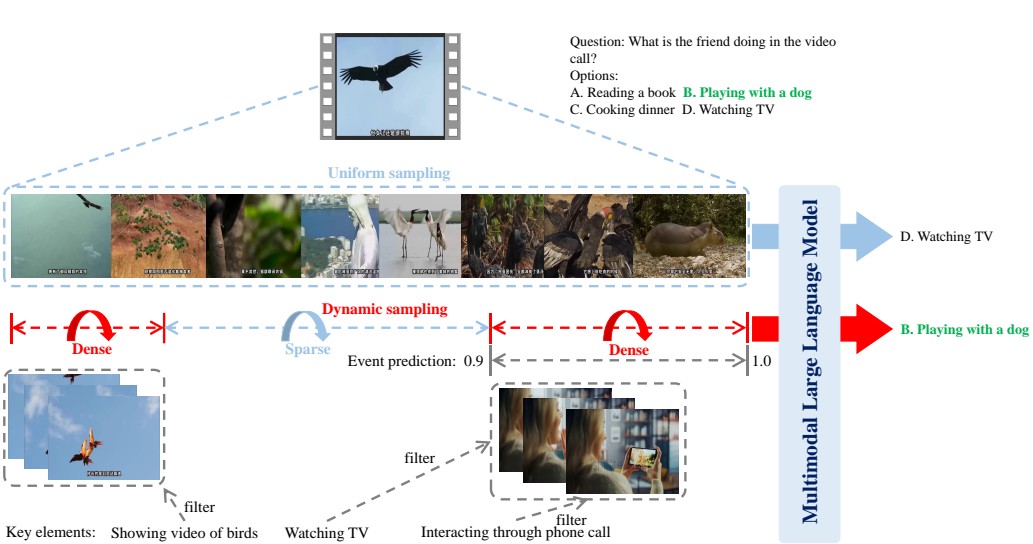

Figure 3: An inference example of the Needle Question Answering task.

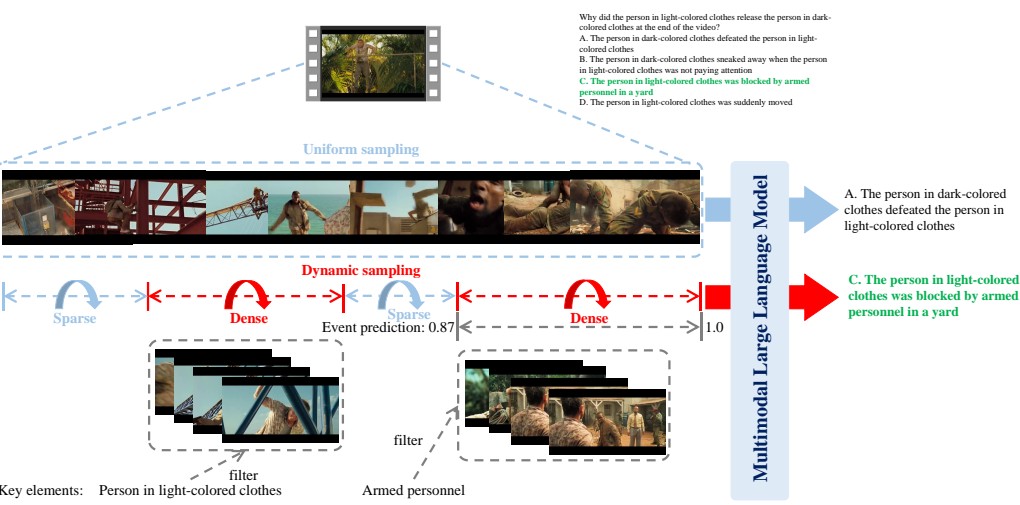

Figure 4: An inference example of the Plot Question Answering task.

