# OpenReview forum: "Reinforcement Learning for Versatile Video Reasoning Capabilities in Base Multimodal LLMs"
_ICLR.cc/2026/Conference — ICLR 2026 Conference Withdrawn Submission_

### Official Review · Reviewer_H3Vp · 2025-10-18

**Soundness:** 2
**Presentation:** 1
**Contribution:** 2
**Rating:** 2
**Confidence:** 5

**Summary:**

In order to address the current limitation of relying on linguistic reasoning of videos, authors propose: (1) 2-staged training method with 2 datasets. Training method performs a cold start with 3 tasks, and GRPO-based training with 2 tasks (2) an inference method aware of key elements and events by performing both dense and sparse sampling. This enables visual reasoning and authors demonstrated the effectiveness with 7 benchmarks.

**Strengths:**

1. Conceptual contribution: The main motivation of the paper of mitigating the sole reliance of linguistic reasoning is very solid. I also believe that relying on one modality is not a good choice, the community should focus on developing reasoning on both modalities.
2. The proposed methods achieve benchmark improvements using only 8k training samples, where existing methods require a larger scale of dataset (165k, 260k pairs for Video-R1).

**Weaknesses:**

1. Writing and readability
- The logic flow is unclear, especially in the intro. e.g., it is unclear what “However, …” in line 74 wants to elaborate, referring to a Table located in the appendix in the introduction distracts the readability (line 83).
- A brief explanation of Cold Start would enhance the readability.
- Multiple notation inconsistency. For example, “MLLM” is declared more than one time and usage is inconsistent (line 243).
- Section 3.1 looks redundant. It only has 3 sentences, and Eq.1 and 5 are identical, only substituted $q$ by $m$. Moving Section 3.1 to appendix and including more details on methods and experimental results would improve the paper.
2. Data curation
- Curating a dataset with a unified format is concerning. [1]. Also, why should we use specific prefixes for each task?
- I am not against using proprietary models for automated annotations, but I believe there should be some careful filtering or verification process for key element labeling in the main script [2].
- Adding actual system prompt and query at Figure 1 will be much more helpful for readers to understand the tasks.
- Would be better if there’s a figure for data curation of 5k video pairs. Also, descriptions on how to generate the data are insufficient.
3. GRPO design
- More details of IoU and Format Reward should be elaborated. Is FormatReward = 1 if the model includes both \<event_start\> and \<event_end\> and 0 otherwise?
- Do we need any normalizations for IoU rewards? Don’t we need to balance three rewards with hyperparameters?
4. Experiments and analyses
- No qualitative results are reported.
- The experiments are only performed with Qwen2-VL-7B. We are curious how well it performs on other base MLLMs as well.

I will increase the score once current concerns are addressed.

**Questions:**

(major)
1. Notation
- The model is not generating the instruction tokens. So shouldn’t it be $X_{instruct}$ rather than $X_{instruct, <i}$ in Eq. 4?
2. Sec 3.2
- How does using trainable parameters <|event_start|> and <|event_end|> help to generate stable predictions? Explanation of the effects with trainable parameters and justification is required.
- How big is the performance gap between adopting pre-trained models and post-trained SFT models?
- Exactly which bias the post-trained SFT model possesses? I believe there is a way to estimate this bias. For example, if the authors are indicating the length of LLM responses, numbers of generated tokens or words can be compared to verify this.
3. Experiments and analysis
- Are there additional impacts of the proposed training and inference on long videos? In other words, any more findings specifically on LongVB, LVBench?
- How can the authors argue that the proposed method is “stable” and “efficient” other than the number of the trainset? I believe a graph of the training curve [3] or the mean and standard deviation across multiple runs [4] should be reported for stability. This paper lacks validation of training efficiency. Also, FLOPs or latency comparison for regular inference vs proposed inference looks necessary. The proposed inference requires additional forwards of external modules (e.g., text encoder, visual encoders, additional generation calls), it does not look efficient.
(minor)
- Is Eq. 5 necessary in this presentation even though we have Eq. 2 and 3?

References

[1] Liang et al. Exploring format consistency for instruction tuning. TMLR 2024.

[2] Li et al. Vidhalluc: evaluating temporal hallucinations in multimodal large language models for video understanding. CVPR 2025.

[3] Lin et al. Rho-1: not all tokens are what you need. NeurIPS 2024.

[4] Leng et al. Mitigating object hallucinations in large vision-language models through visual contrastive decoding. CVPR 2024.

---

### Official Review · Reviewer_ij64 · 2025-10-31

**Soundness:** 2
**Presentation:** 2
**Contribution:** 2
**Rating:** 2
**Confidence:** 5

**Summary:**

This paper proposes VideoReasoner, an efficient framework that introduces versatile video reasoning capabilities for video LLMs. It combines multi-task reinforcement learning and adaptive frame sampling during evaluation, significantly improves the final performance on various benchmarks and outperforms existing GRPO-only video reasoning models.

**Strengths:**

- The performance of VideoReasoner significantly outperforms the base model as well as some prior works focusing on video reinforcement learning.
- VideoReasoner acquires more limited computational resources compared to other video reasoning methods, making it more reproducible.

**Weaknesses:**

- The proposed keyframe level and event level frame sampling method lacks enough comparison to existing methods like Koala [1], Frame Voyager [2], KeyVideoLLM [3], and Logic-in-Frames [4]. Also, it shows limited advantage and novelty compared to these methods. I believe that using these lightweight modules as the key frame sampling method during inference should also work.
- Regarding the weak connection between the main reasoning part and the frame sampling part, the results in Table 5 show that much performance comes from the frame sampling method, and Table 5 demonstrates that there is only a limited gain for VideoReasoner on Instruct models. If the frame sampling is removed (since other models can also easily adapt to these sampling methods), I guess the final performance might drop just like Video-R1.
- The proposed method only applies well to base models, which limits the final performance. Also, recent models like Qwen 2.5-VL and Qwen 3-VL do not provide base models anymore, which raises my concern about the generalizability of the method for future stronger models.
- Since H20 GPU has enough VRAM, the performance on a higher frame rate should also be revealed, which is also a general manner in recent models, like Qwen 2.5-VL generally samples 768 frames. In this case, the frame sampling method might not work well.
- The figures are not presented well. Figure 1 uses the same color for too many blocks, which makes it difficult for the readers to understand. Figure 2 should be reformatted to reduce confusion.

[1] Tan et. al, Koala: Key frame-conditioned long video-LLM, in Proc. CVPR 2024
[2] Yu et. al, Frame Voyager: Learning to Query Frames for Video Large Language Models, in Proc. ICLR 2025
[3] Liang et. al, KeyVideoLLM: Towards Large-scale Video Keyframe Selection, in CoRR 2024
[4] Guo et. al, Logic-in-Frames: Dynamic Keyframe Search via Visual Semantic-Logical Verification for Long Video Understanding, in Proc. NeurIPS 2025

**Questions:**

- Could the authors provide a more detailed comparison to other frame sampling methods and provide more explanation of the connection between the frame sampling part and the reasoning part?
- Could the authors provide the reasoning-only performance on Instruct models for fair comparison to existing methods?
- Could the authors provide results on models with a larger frame number to further prove the effectiveness of the frame sampling strategy?

---

### Official Review · Reviewer_4rtZ · 2025-10-31

**Soundness:** 2
**Presentation:** 2
**Contribution:** 3
**Rating:** 4
**Confidence:** 4

**Summary:**

This paper introduces VideoReasoner, a novel framework designed to address the issues of hallucination and poor reasoning in Multimodal Large Language Models (MLLMs) for complex video understanding. Its core strategy is to build and enhance versatile reasoning capabilities on a more malleable Base Model, diverging from the common practice of applying Reinforcement Learning (RL) to already fine-tuned Instruct Models. The framework employs a two-stage training process, featuring a "multi-task cold start" (SFT) and a subsequent multi-task RL phase, to develop the model's multi-perspective reasoning across events and keyframes. The authors claim this approach is highly data-efficient, using only 8k samples to achieve performance that significantly surpasses the base model and even outperforms state-of-the-art Instruct Models on several public benchmarks.

**Strengths:**

1. The paper's core hypothesis—that training complex reasoning is more effective on Base Models than on Instruct Models—is insightful, novel, and strongly supported by the experiments.

2. The framework is highly data-efficient, effectively cultivating complex video reasoning capabilities with only a small amount of data (8k), which showcases its significant potential for low-cost model adaptation.

**Weaknesses:**

1. The paper critically lacks any analysis of inference efficiency. The proposed inference pipeline, as detailed in Figure 2 of the Method section, clearly involves a complex, multi-step process requiring at least three separate forward passes through the model (for event reasoning, keyframe reasoning, and final response generation), plus additional post-processing. However, Section 4 (Experiment) provides no data whatsoever on inference latency or throughput.

2. The framework's generalizability is questionable as all experiments were exclusively conducted on the Qwen2-VL model family. It remains unclear if the core findings would hold for other prominent MLLM architectures.

**Questions:**

1. Regarding the Reward Function Design: In the multi-task RL stage, the reward function $r(o)$ (Equation 6) is a sum of three different components ($r_{\text{IoU}}$, $r_{\text{form}}$, and $r_{\text{acc}}$). These components might have very different scales (e.g., $r_{\text{IoU}}$ is continuous in $[0, 1]$, while $r_{\text{acc}}$ is discrete in $\{0, 1\}$). Were any normalization or weighting schemes applied to these rewards? If not, did you observe any optimization imbalance where the model might prioritize the reward signal with a larger numerical range?

2. The ablation study in Table 4 demonstrates that combining the "Event" and "Keyframe" reasoning paths yields the best results. I would like to ask for a further clarification: what would the performance be if the fully trained (SFT+RL) model was used for direct, single-pass inference without the complex pipeline from Figure 2? I would like the author to directly quantify the exact performance gain attributable to the costly, multi-step inference process.

---

### Official Review · Reviewer_xTxx · 2025-11-01

**Soundness:** 3
**Presentation:** 2
**Contribution:** 3
**Rating:** 6
**Confidence:** 4

**Summary:**

This paper proposes VideoReasoner, a two-stage training and inference framework to improve video reasoning for base multimodal LLMs. The pipeline is as follows:
(1) a multi-task supervised fine-tuning “cold start” (video QA, event grounding, key-element generation),
(2) a multi-task Group Relative Policy Optimization (GRPO) reinforcement learning stage with task-specific rewards (IoU, format, accuracy), (3) an inference pipeline that runs event grounding and key-element/keyframe reasoning in parallel and then dynamically samples (dense on key intervals, sparse elsewhere) to produce final answers.
The authors demonstrate empirical gains on seven public video benchmarks using Qwen2-VL-7B-Base with only ~3k cold-start and 5k RL samples, showing consistent improvements and in several cases outperforming instruct-tuned baselines.

**Strengths:**

The paper demonstrates several notable strengths:
(1) Well-designed multi-task RL framework. The use of task-specific prefixes to generate multiple outputs from the same (video, query) input, combined with carefully designed task-dependent rewards, is both elegant and data-efficient.   This multi-task GRPO setup effectively aligns different reasoning skills within a unified optimization framework.
(2) Effective multimodal reasoning and inference strategy. The integration of event grounding and keyframe extraction enables dynamic dense–sparse frame sampling during inference, which is a practical and efficient solution for reasoning over long or complex videos. This design significantly reduces computational cost while maintaining temporal understanding.
(3) Strong data efficiency. The proposed framework achieves remarkable performance using only a small-scale training setup—approximately 3k samples for supervised fine-tuning and 5k for reinforcement learning. Such data efficiency highlights the method’s practical value for extending video reasoning capabilities without requiring massive annotation budgets.

**Weaknesses:**

(1) The paper directly sums three reward components (IoU, format, and accuracy) but lacks a systematic analysis of their relative weights, normalization strategies, and potential conflicts between tasks.  More experiments on reward weighting, per-task learning curves, and training stability would substantially strengthen the claims.
(2) In Table 5, the Qwen2-VL-7B-Instruct + RL model achieves only 38.3 on the M-Avg column of MLVU, showing a large drop compared with the baseline (60+).  The authors should clarify whether this performance degradation is expected, provide an explanation (e.g., task interference, reward imbalance), or verify if there was a reporting error.
(3) The paper omits several crucial implementation details, including GRPO hyperparameters (group size G, ϵ, β), number of update steps, learning rate schedule, and overall compute cost.  A reproducibility table summarizing these parameters and GPU-hour statistics would make the work more transparent.
(4) The key-element annotations used in the cold-start phase rely on a proprietary model.  Using or benchmarking against an open-source alternative would make the approach more credible and reproducible for the broader community.
(5) While the proposed inference pipeline (Figure 2) is conceptually elegant, it appears computationally expensive—it requires at least two full MLLM forward passes plus similarity-based keyframe retrieval. The paper highlights training efficiency but overlooks the potential inference latency and cost. This trade-off should be explicitly acknowledged and, ideally, quantitatively analyzed (e.g., latency in ms or FLOPs relative to baseline).

**Questions:**

(1) The Qwen2-VL-7B-Instruct + RL model achieves only 38.3 on MLVU (M-Avg), which is significantly lower than the baseline (Table 5).  Could the authors explain the reason for this degradation?  Was it due to task interference, suboptimal reward scaling, or possibly a reporting issue?  Clarifying this would help readers understand the stability of your RL process.
(2) The inference pipeline (Fig. 2) seems to require multiple forward passes and similarity-based retrieval.  Could the authors quantify the additional computational overhead (latency, FLOPs, or throughput) compared with a standard single-pass baseline?  Are there optimizations to make this process more efficient in practice?
(3) The paper acknowledges a performance gap when applying the proposed RL method to the Instruct model, noting that further work is needed to bridge this gap. Could the authors elaborate on the underlying reasons for this degradation? Any diagnostic analysis or ablation would help clarify why the Instruct model performs notably worse than the Base one.
(4) Please include a complete list of GRPO training hyperparameters (e.g., group size G, clipping parameter ϵ, β, batch size, learning rate, number of updates) and report the total compute cost (e.g., GPU hours).  This information is essential for ensuring reproducibility and fair comparison.
(5) Since the cold-start data relies on a proprietary model for key-element generation, can you release the generated annotations or provide an open-source alternative? If not, could you at least test whether replacing this component with a public model yields similar trends?

---

### Official Review · Reviewer_gzCT · 2025-11-04

**Soundness:** 3
**Presentation:** 2
**Contribution:** 2
**Rating:** 2
**Confidence:** 4

**Summary:**

This paper introduces VideoReasoner, a framework designed to efficiently enhance the video reasoning capabilities of base Multimodal Large Language Models (MLLMs). The authors identify two primary weaknesses in existing Reinforcement Learning (RL) approaches for video: 1) unstable and costly training, and 2) over-reliance on purely linguistic reasoning, which can lead to factual inaccuracies (hallucinations). To address this, the paper introduces a 2-stage training method on a _base_ MLLM:

1. Multi-task Cold Start: An SFT phase on a small dataset (3k samples) that teaches the model three distinct skills using specific prefixes: direct question answering, event grounding (locating an event in time), and key element extraction.
2. Multi-task Reinforcement Learning: An RL phase using GRPO on another small dataset (5k samples) with 3 RL rewards: two for correct event grounding (IoU for predicting event ranges and format rewards) and accurate question answering (accuracy reward).

Finally, the paper introduces an efficient inference pipeline to take advantage of this training, which uses the event grounding capability to identify regions to densely sample, leaving the remaining areas to be sparsely sampled. This allows the process overall to be more efficient in terms of informative number of frames. The numbers improve over the base Qwen2-VL, but still somewhat fall short of Qwen-2.5-VL-Instruct.

**Strengths:**

1. **Good data and training efficiency**: The most compelling strength is the high sample efficiency. Achieving strong performance with only 3k SFT and 5k RL samples stands in stark contrast to prior work like Video-R1, which uses hundreds of thousands of examples. This makes the method more accessible and practical, significantly lowering the barrier to training capable video reasoning models.

2. **Motivated Multimodal Reasoning and Inference Pipeline**: The paper moves beyond purely textual chain-of-thought by incorporating multimodal elements (events, keyframes) directly into the reasoning process. The proposed inference pipeline, which uses the model's own preliminary analysis to dynamically focus its attention, is an interesting way to tackle the "needle in a haystack" problem in long videos, albeit somewhat done before.

3. **Strong Empirical Results and Comprehensive Ablations**: The method shows consistent and significant improvements over the Qwen2-VL-7B-Base baseline across seven different benchmarks (Table 1). The ablation studies in Table 4 are particularly effective, clearly demonstrating that each component (cold start, RL with both QA & Temporal Grounding, and the combined inference strategy) contributes positively to the final performance.

**Weaknesses:**

1. **Inconsistent comparison of results**: The key results in Table 1 are quite confusingly laid out. As mentioned in the text, doing the pipeline on instruct-tuned models doesn't seem as effective, which is fair. Also there is no released Qwen-2.5-VL-Base model as far as I can tell. However, why is the comparison of GRPO to Qwen-2.5-VL-Instruct instead of Qwen-2-VL-Instruct? Also is GRPO necessary for this process? A pure SFT/pure GRPO baseline would be a useful datapoint for the community to understand where the benefit is coming from (data points, data structure, or the actual algorithm). In fact, my main takeaway from Table 5 is perhaps that GRPO for post-trained models is bugged, or perhaps this implementation of it was? The cold start works quite well (Q2-VL-7B-I largely better than Q2-VL-7B-B), which makes it hard to believe the RL is adding anything here. Plus, if the RL doesn't work on instruct tuned models, wouldn't that render this method largely incompatible anyways with modern frameworks, as we would like to end on RL anyways (unless you can instruct after RL)?

1. **Lack of Inference Latency Analysis**: The framework is presented as "efficient," but this only refers to training. The inference process involves at least two passes through the model (parallel generation of events/elements, followed by the final QA) plus intermediate processing and frame sampling. This is computationally more expensive and will have higher latency than a standard single-pass approach. The complete omission of this detail is a notable flaw. See Questions for followup.

1. **Vague Description of Key Method**: It's not clear to me exactly how VideoReasoner works (L350-354). The paper mentions using text and visual encoders to find frames with "high similarity," but the details of this similarity metric and the retrieval process are absent. Is this just cosine similarity lookup? How is the frame merging done afterwards?

1. **Potential Dataset Leakage**: Table 3's results on Charades-ST seems to be very strong, but upon looking into this further, it seems one of the datasets, TALL, which is used to derive the 3k cold start phase, is largely based on Charades-ST itself. Therefore this is not a fair evaluation anymore between the two methods, if there is data overlap here. Clarification to this point is important.

**Questions:**

1. In L372 you say that you sample 64 frames per video for both training stages and inference, but you're also doing dense/sparse sampling of frames based on the events. How exactly is this broken down?
1. What is the actual throughput of the method? Can you compare the inference pipeline . It's similar to SeViLA [1] and more recent works which do frame sampling, but SeViLA actually runs at a comparable speed because the frame selection model (BLIP) is relatively lightweight and limits the processing you'd otherwise do. Here, you have to process both the densely sampled and sparse sampled frames, and you're using a full fledged MLLM twice. Given you're also processing the video fully at the beginning, I don't see how this is nearly twice the time it takes to do this with a normal query. In this case, it's not a fair comparison to the base Qwen method, so I would recommend compute matching either at higher frames or applying VideoReasoner to Qwen as well (or other similar method, which won't do as well since it's not trained for it).
1. In general, it's not clear that GRPO is necessary, or at least I'd like a clear experiment showing its benefit on a data matched setting over SFT.

Overall, while the proposed method is technically interesting, there are many outstanding areas of concern with how real the results may be. My current assessment of the paper is a reject, but I would be happy to raise my score following increased discussion with the authors and addressing of my points above.


[1] Yu, Shoubin, et al. "Self-chained image-language model for video localization and question answering." Advances in Neural Information Processing Systems 36 (2023): 76749-76771.

---

### Note · Authors · 2025-11-14

I have read and agree with the venue's withdrawal policy on behalf of myself and my co-authors.